# MsIFT: Multi-Source Image Fusion Transformer

**Xin Zhang** [1,2] , **Hangzhi Jiang** [1,2], **Nuo Xu** [1,2], **Lei Ni** [3], **Chunlei Huo** [1,2,*] **and Chunhong Pan** [1,2]

1 National Laboratory of Pattern Recognition, Institute of Automation, Chinese Academy of Sciences, Beijing 100190, China

2 School of Artificial Intelligence, University of Chinese Academy of Sciences, Beijing 101408, China

3 Beijing Institute of Remote Sensing, Beijing 100085, China

* Correspondence: clhuo@nlpr.ia.ac.cn

**Abstract:** Multi-source image fusion is very important for improving image representation ability since its essence relies on the complementarity between multi-source information. However, feature-level image fusion methods based on the convolution neural network are impacted by the spatial misalignment between image pairs, which leads to the semantic bias in merging features and destroys the representation ability of the region-of-interests. In this paper, a novel multi-source image fusion transformer (MsIFT) is proposed. Due to the inherent global attention mechanism of the transformer, the MsIFT has non-local fusion receptive fields, and it is more robust to spatial misalignment. Furthermore, multiple classification-based downstream tasks (e.g., pixel-wise classification, image-wise classification and semantic segmentation) are unified in the proposed MsIFT framework, and the fusion module architecture is shared by different tasks. The MsIFT achieved state-of-the-art performances on the image-wise classification dataset VAIS, semantic segmentation dataset SpaceNet 6 and pixel-wise classification dataset GRSS-DFC-2013. The code and trained model are being released upon the publication of the work.

**Keywords:** transformer; multi-source image fusion; non-local

## 1. Introduction

Due to different imaging mechanisms between multi-source remote sensing images, accurate pixel-wise registration is difficult, and the spatial inconsistence as well as the resulted feature semantic bias will be further propagated to the subsequent fusion procedure. As illustrated in Figure 1a, there are large displacements between the SAR image and the optical image (e.g., the building marked by the yellow dashed box and the corresponding building marked by the white dashed box), even when the SAR image and the optical image are aligned carefully. When features at the same position are merged, the semantic bias will produce noise and weaken the discriminative ability of the features, and the performance of downstream tasks based on multi-source images fusion will thus be impacted.

With the development of deep learning in recent years, deep neural networks (DNN) (e.g., convolution neural network (CNN) [1], recurrent neural network (RNN) [2], long short-term memory (LSTM) [3] and the capsule network [4]) have been introduced to multi-source images fusion. In the literature, CNN is the most widely used network, and it dramatically improves the representation ability of multi-source images. A variety of novel multi-source image fusion methods have been proposed within the CNN framework. However, the inherent local inductive bias of CNN limits the receptive field of features. As shown in Figure 1b, the feature noise caused by semantic bias is hard to be alleviated, even by increasing CNN layers, where the number of network layers in the backbone is 50. In short, semantic bias is the key bottleneck of multi-source image fusion.

To address the above difficulty, a multi-source remote sensing image fusion method with the global receptive field is proposed in this paper, which is named after the MsIFT

(multisource image fusion transformer). Since the transformer was introduced into computer vision, promising potentials have been shown in many visual tasks such as recognition [5], detection [6], segmentation [7] and tracking [8]. The reason lies in its ability to capture long-range dependence by the global receptive field. In this context, we construct a non-local feature extraction and feature fusion module based on the transformer. As shown in Figure 1c,d, self-attention and cross-attention are essential components of the above two modules, respectively. The former is to find the aggregatable features (brighter area) from the local source image for query point, and the latter is to find the aggregatable features (brighter area) from the other source image. In selecting features for aggregation, the object query point tends to find features with the same or globally related semantics, and this mechanism is the key of the MsIFT to overcome the feature semantic bias. As shown in Figure 1d, when the SAR image is fused with the optical image, the SAR image features will be aggregated with the features on the highlighted area in the optical image, rather than simply aggregating the optical image features at the same spatial location. As an example, the feature points of the building location are marked in yellow; the MsIFT fuses the building features in the SAR image with the building area features in the optical image. In a word, the MsIFT can reliably merge features through the globality of transformers even if the semantics of multi-source images are not aligned.

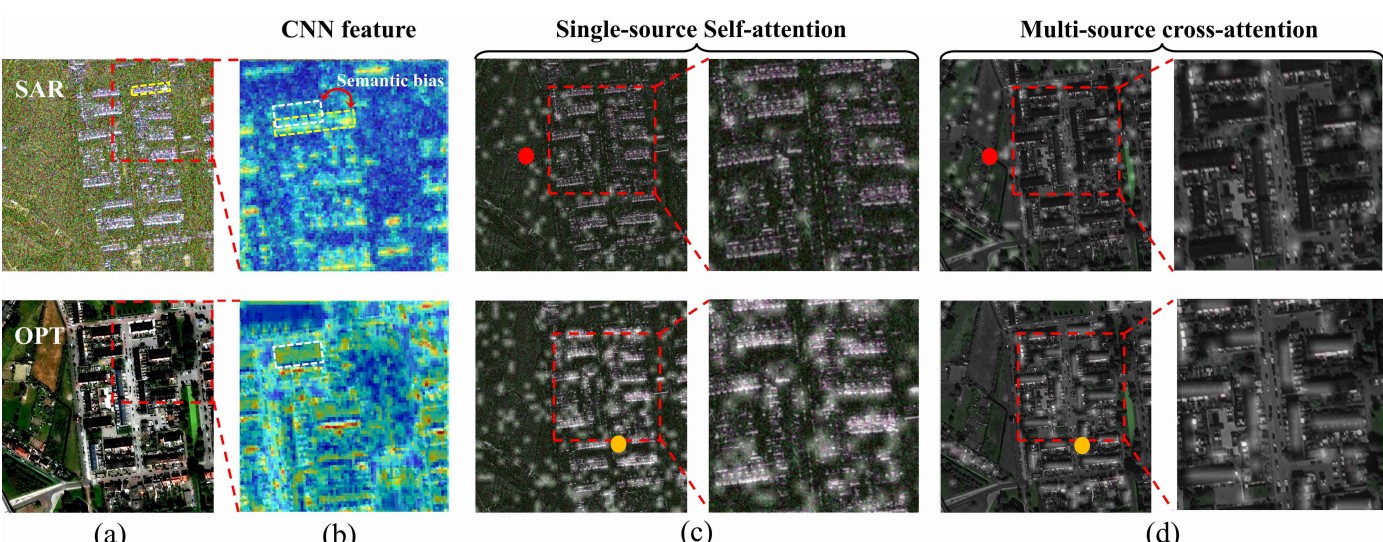

**Figure 1.** Limitation of direct pixel-wise fusion and advantages of the MsIFT in addressing misaligement by global receptive fields. (**a**) Synthetic aperture radar (SAR) image and optical (OPT) image. (**b**) CNN features of multi-source images. (**c**) Attention maps of the picked spatial points from SAR image during feature extraction. (**d**) Attention maps of the picked spatial points from the optical image during feature fusion. The brighter region indicates the area that the query points pay more attention to. As shown in (**c**,**d**), the **red** background query points pay more attention to the background region, while the **yellow** object query points pay more attention to the objects' region. The query will aggregate the features on the attention region at the feature extraction and fusion stage. Therefore, the MsIFT is powerful in overcoming the semantic bias caused by the misaligned multi-source images.

In addition, multiple downstream tasks (e.g., **pixel-wise classification** (PWC), **image-wise classification** (IWC) and **semantic segmentation** (SS)) based on multi-source image fusion are integrated into the MsIFT framework, and multiple downstream tasks share the same fusion network structure. The contributions of this paper are as follows:

- A multi-source image fusion method with the global receptive field is proposed. The non-locality of the transformer is helpful for overcoming the feature semantic bias caused by semantic misalignment between multi-source images.

- Different feature extractor and task predictor networks are proposed and unified for three classification-based downstream tasks, and the MsIFT can be uniformly used for pixel-wise classification, image-wise classification and semantic segmentation.
- The proposed MsIFT improved the classification performance based on fusion and achieved state-of-the-art (SOTA) performances on the VAIS dataset [9], SpaceNet 6 dataset [10] and GRSS-DFC-2013 dataset [11].

## 2. Related Works

### 2.1. DNN-Based Multi-Source Image Classification

Aziz et al. [12] proposed a dual-stream CNN for multi-source image classification, where multi-source features obtained by the dual-stream network are fed to the fully connected (FC) layer for feature-level fusion. Different from [12], Qiu et al. [2] presented a multi-layer block RNN for learning feature representation from CNN features. Hierarchical features from different sources are merged through cascade operation and sent to the support vector machine (SVM) for classification. Santos et al. [13] proposed a decision-level fusion method based on a dynamic probability model. After enhancing visible images, additional sensor data were simulated to improve the classification accuracy. Xu et al. [1] proposed a two-branch CNN network, i.e., dual tunnel CNN and cascade network block. The former is to learn spatial and spectral features of multi-channel images (e.g., hyperspectral images, HSIs), and the latter is to extract spatial features of single-channel images (e.g., LiDAR data). Hong et al. [14] proposed a multi-modal learning framework for pixel-wise classification and compared five different fusion modules. Zhu et al. [15] proposed a triple branch fusion network for merging panchromatic (PAN) and multi-spectral (MS) remote sensing images. The spatial-spectral features extracted by the fusion branch enhance the spatial features of the PAN branch and the spectral features of MS branch.

However, object representations between multi-source images exhibit significant differences due to the heterogeneous nature of different sensors. Multi-source information cannot be equally represented [16]: simple stacking, adding and concatenation manner may cause feature redundancy. Li et al. [17] proposed an asymmetric feature fusion network for multi-source remote sensing image classification. To eliminate the unbalance of multi-source data information, a sparse constraint method and feature calibration module are proposed to reduce the redundant information. A weight-shared CNN is utilized to reduce the complexity of the multi-source feature extraction network. Zhang et al. [18] proposed an interleaving perception convolutional neural network for integrating the heterogeneous information of multi-source data. Mohla et al. [19] proposed a attention-based feature fusion method. The attention mask is obtained from one modality feature to highlight the features in the other modality feature. Peng et al. [20] proposed a spatio-temporal–spectral fusion framework based on semi-coupled sparse tensor factorization. Li et al. [4] introduced the maximum correntropy criterion into the capsule network to address the noise and outliers problems in HSIs. Furthermore, a dual-channel capsule network framework was proposed and applied to the pixel-wise fusion-based classification. Compared with CNN and capsule networks, LSTM is more powerful in modeling the long-range dependence along the spectral dimension. Hu et al. [3] introduced LSTM into HSIs classification and proposed a new deep model ConvLSTM to extract more discriminative spatial–spectral features. Heng Chao Li et al. [21] extended ConvLSTM to pixel-wise fusion-based classification. For HSIs and LiDAR data, spatial and spectral attention modules are utilized to extract spatial–spectral features, respectively. The multi-scale residual attention module is designed to learn multi-scale features. At the fusion stage, a three-level fusion strategy is proposed for feature-level fusion.

Due to the limited training data of multi-source remote sensing images, those above-mentioned supervised learning-based convolutional neural networks lack generalization ability. Some multi-source remote sensing image feature fusion methods based on unsupervised and semi-supervised convolutional neural networks (e.g., encoder–decoder-like models) have been proposed. Hong et al. [22] proposed a semi-supervised cross-modal

classification network to transfer the discriminative information from small-scale source data into the classification task using large-scale source data. Zhang et al. [23] designed an unsupervised network with multi-layer stacked auto-encoder to learn the translation function between two image domains. The two source data were used as the input and output of the auto-encoder network, respectively. Additionally, the underlying representation can be regarded as the fusion feature obtained from the two source data.

To solve the problem caused by feature imbalance, decision-level fusion methods for multi-source images classification are also proposed. Huang et al. [24] proposed the first semantic segmentation network of multi-source remote sensing images based on decision-level fusion. A fully convolution network (FCN) is selected for the segmentation network, and the FCN is composed of an encoder and a decoder. The former is used to extract image features, and the latter is used to decode and classify features. Finally, a voting mechanism is chosen for decision-level fusion. Liao et al. [25] ensemble feature fusion and decision fusion together for multi-sensor data classification. The spectral, spatial, elevation and graph-fused features were fed into the SVM classifier. The final classification map was obtained by fusing the four classification results through the voting mechanism. Hang et al. [26] proposed a weighted summation decision-level fusion method. Especially, the weights are determined by the classification performance of the classifier during training.

### 2.2. Transformer in Computer Vision

Despite the advantages of local perception ability inherent in CNN, it limits the receptive field of the network, and it often ignores the semantic relationship between objects and the context relationship between objects and background. As a consequence, long-range dependence should be considered in feature learning. The attention mechanism makes the transformer suitable for modeling the long-range dependence. Inspired by [27], Dosovitskiy et al. [5] used the transformer instead of CNN for the backbone network and proposed the vision transformer (ViT) for image classification. ViT divides the image into patches and treats each patch as a word. Then, the patch embeddings are sent to the encoder for feature extraction. In the encoding process, with the help of the self-attention layer, each patch feature has a global receptive field during feature extraction. ViT demonstrates that transformer-like architectures have the ability to compete with state-of-the-art CNNs (e.g., ResNet [28]), which has dramatically stimulated researchers' enthusiasm for introducing the transformer into the computer vision world. Chen et al. [29] extended the transformer to low-level vision tasks (i.e., deraining, denoising and super-resolution). Multiple head and tail networks are designed for different tasks, and different tasks share the same transformer. After fine-tuning, the model pre-trained on ImageNet beat the current SOTA model on multiple vision tasks. Carion et al. [6] proposed a novel object detection framework based on the transformer, DETR. It is composed of an encoder and a decoder. The encoder is used to extract features from the image. In the decoder, a set of query embeddings are used to represent the anchor boxes. The embeddings perform cross attention on global features, and the final features are used for prediction. Zheng et al. [7] proposed a transformer-based semantic segmentation framework, SETR. Similarly, each patch embedding aggregates the context information by self-attention, and the learned features have the global receptive field. Hu et al. [30] proposed a transformer-based instance segmentation framework, ISTR. In the decoder, the cross attention operation is performed by the learnable query boxes. Three different head networks predict category, location and mask for each region of interest. Chen et al. [31] proposed a transformer-based change detection framework, BiT. In the encoder, the features of image pairs are queried and encoded with each other. In the decoder, the encoded features are queried and decoded with the corresponding features. The dual-stream features are sent to the prediction head to obtain the change map.

The success of the transformer in computer vision depends on the attention mechanism in modeling the long-range dependence. Unlike the above images' fusion work, in this paper, the transformer is used to enhance the global interaction ability between features

in a single image. In the other words, the global interaction ability between image pairs is improved to overcome the semantic deviation caused by local fusion. Different from traditional methods based on the transformer, we extend the transformer in a new research field, i.e., multi-source remote sensing image fusion and classification, and propose a novel multi-source fusion framework to deal with a variety of classification tasks.

## 3. Method

The MsIFT is a multi-source fusion framework, by which it unifies multiple classification-based downstream tasks. In this section, the MsIFT is elaborated in detail.

### 3.1. Problem Formulation

Let $\mathcal{I}_1 \in \mathbb{R}^{H \times W \times C_1}$ and $\mathcal{I}_2 \in \mathbb{R}^{H \times W \times C_2}$ be an image pair from different sources with $C_1$ and $C_2$ channels, respectively. The goal of the MsIFT is to perform non-local feature-level fusion on the image pair, and fused features are directly used for the *i*-th downstream task:

$$\tilde{f}_{p_i} = \mathcal{F}_{t_i}(\text{Fusion}(\mathcal{F}_{h_i}^1(\mathcal{I}_1), \mathcal{F}_{h_i}^2(\mathcal{I}_2))), \tag{1}$$

where $\mathcal{F}_h$ represents CNN-based feature extractor. Fusion is the non-local feature-level fusion operator, and $\mathcal{F}_t$ is the prediction network.

The MsIFT generates the score map $\tilde{f}_{p_i} \in [0,1]^{h \times w \times C}$ for task $T_i$, $C$ denotes the number of categories and $h \times w$ is the number of samples to be classified. $\tilde{f}_{p_i}$ represents the classification probability of the sample $p_i$, i.e., $\sum_{c=1}^{C} \tilde{f}_{p_i}^{(j,k,c)} = 1, j = 1, \ldots, h$ and $k = 1, \ldots, w$. Finally, the label of the sample $p_i$ is obtained by:

$$\tilde{c}_{j,k} = \arg\max_c \tilde{f}_{p_i}^{(j,k,c)}. \tag{2}$$

### 3.2. MsIFT Architecture

As shown in Figure 2, the MsIFT consists of CNN feature extractor, feature fusion transformer and the task predictor. In the MsIFT, image pairs are encoded twice before feature fusion, i.e., local-based and global-based feature aggregation are implemented by CNN feature extractor $\mathcal{F}_h$ and fusion transformer encoder, respectively. The decoder in the fusion transformer conducts non-local feature aggregation from different sources. The task predictor generates the final labels for each task.

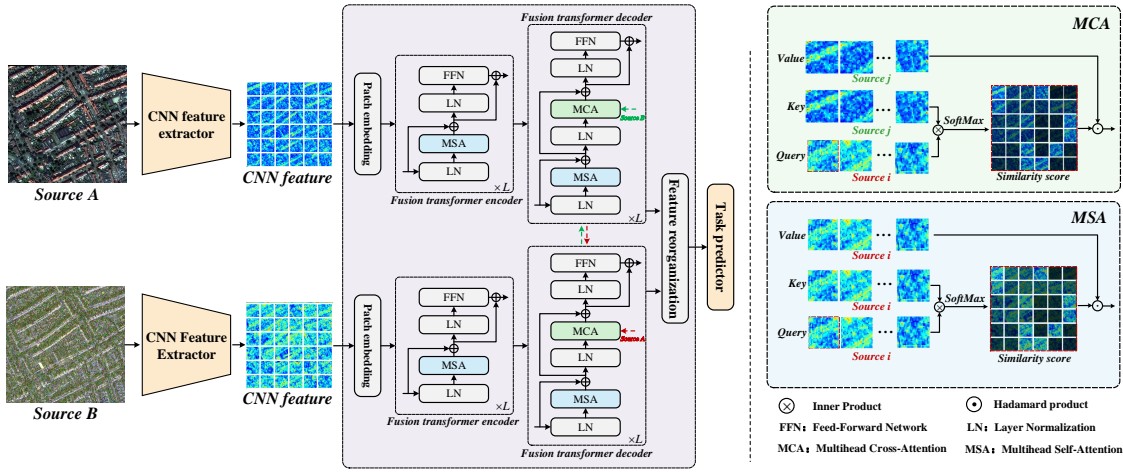

**Figure 2.** Overview of the proposed multi-source image fusion transformer (MsIFT). The MsIFT consists of the CNN feature extractor, feature fusion transformer and task predictor. The input images are first fed into the CNN feature extractor to obtain the local visual feature, the feature fusion transformer with encoder–decoder style structure is conducted to perform the global feature-level fusion. The prediction result is produced by the task predictor.

### 3.2.1. CNN Feature Extractor

Considering the differences between the downstream tasks, CNN feature extractors (CFEs) are designed separately for each task. CNN introduces the local induction bias to the network to extract shallow features with translation invariance. The image from different sources uses the same network architecture, but parameters are not shared.

Figure 3 shows CFEs of three downstream tasks. The downsampling factors of the output feature maps are 2, 1 and 8, respectively. Specifically, the input image scale of IwC and PwC is small, so feature maps with small downsampling factors are used as the output to preserve the spatial details. To better use the global attention mechanism, in the fusion module, the top-down fusion is performed on multi-scale feature maps, and high-resolution feature maps with semantic information are obtained. For semantic segmentation task with a large input size, the feature map with a downscale factor of 8 is used as the output. Consistent with the general semantic segmentation model, dilated convolution is applied on deep features to improve the receptive field of the convolution kernel, and the number of parameters is kept unchanged.

CFE produces the CNN feature $f_h \in \mathbb{R}^{H' \times W' \times C}$ for each source and task.

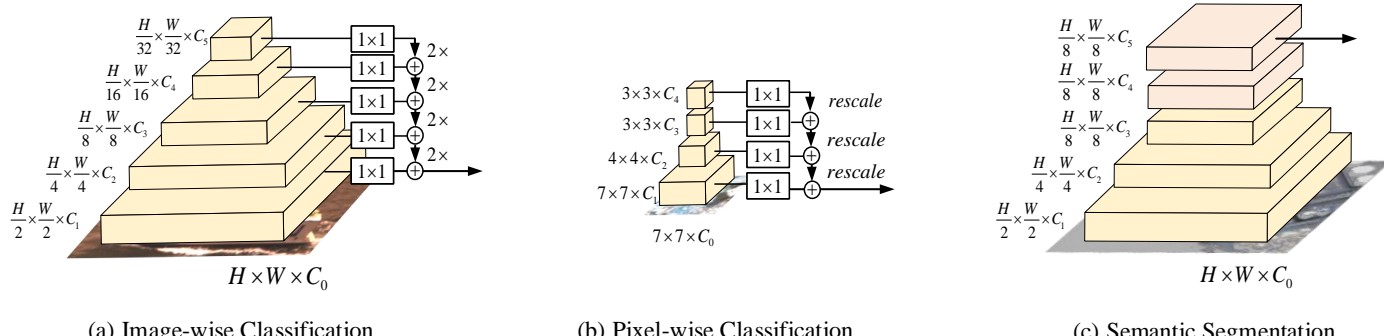

(a) Image-wise Classification      (b) Pixel-wise Classification      (c) Semantic Segmentation

**Figure 3.** CNN feature extractor networks. Different feature extractor networks are proposed for three classification-based downstream tasks. For (**a**) image-wise classification and (**b**) pixel-wise classification of tasks, the top-down fusion is performed on multi-scale feature maps to obtain the high-resolution feature maps with semantic information. For (**c**) semantic segmentation task, dilated convolution is applied on deep features to improve the receptive field of the convolution kernel.

### 3.2.2. Feature Fusion Transformer

**The fusion transformer encoder (FTE).** FTE extracts deep semantic features through the non-local attention mechanism. Specifically, CNN feature $f_h$ is spatially divided into a sequence of 2D patches $\{\mathbf{f}_{h_i}\}_{i=1}^{N}$, where $\mathbf{f}_{h_i} \in \mathbb{R}^{P^2 \times C}$, $P$ is the patch size and $N = \frac{H \times W}{P^2}$ is the number of patches. A linear projection layer $\mathbf{E} \in \mathbb{R}^{(P^2 \cdot C) \times D}$ is trained to map flattened patches to $D$-dimension latent embedding space. To make the network sensitive to the spatial information, a sequence of 1-D learnable position embeddings $\{\mathbf{E}_{p_i}\}_{i=1}^{N}$ are added to the patch embeddings, respectively, where $\mathbf{E}_{p_i} \in \mathbb{R}^{D}$. $\{\mathbf{f}_{h_i}\mathbf{E} + \mathbf{E}_{p_i}\}_{i=1}^{N}$ is sent to the stacked transformer encoder, and each encoder layer consists of multi-head self-attention (MSA) and feed forward network (FFN). Following [5], the skip connection, GELU [32], and layer norm (LN) [33] are used in the encoder layer. The above procedures are formulated as

$$\mathbf{x}_0 = [\mathbf{f}_{h_1}\mathbf{E} + \mathbf{E}_{p_1}; \mathbf{f}_{h_2}\mathbf{E} + \mathbf{E}_{p_2}; \cdots ; \mathbf{f}_{h_N}\mathbf{E} + \mathbf{E}_{p_N}], \tag{3}$$

$$\mathbf{x}'_l = \mathrm{MSA}(\mathrm{LN}(\mathbf{x}_{l-1})) + \mathbf{x}_{l-1}, \tag{4}$$

$$\mathbf{x}_l = \mathrm{FFN}(\mathrm{LN}(\mathbf{x}'_l)) + \mathbf{x}'_l, \qquad l = 1 \ldots L \tag{5}$$

$$[\mathbf{f}_{FTE_1}, \mathbf{f}_{FTE_2}, \ldots, \mathbf{f}_{FTE_N}] = \mathrm{LN}(\mathbf{x}_L). \tag{6}$$

In Equation (4), for the $n$-th patch embedding, the similarities between the $n$-th patch embedding and all other patch embeddings are calculated, and the similarities are used

as the aggregation weights in encoding the $n$-th patch. Noting MSA is a non-local feature extraction process, which is calculated by the following formula:

$$\mathbf{q}_l^i = \text{LN}(\mathbf{x}_{l-1})\mathbf{E}_q, \tag{7}$$

$$\mathbf{k}_l^i = \text{LN}(\mathbf{x}_{l-1})\mathbf{E}_k, \tag{8}$$

$$\mathbf{v}_l^i = \text{LN}(\mathbf{x}_{l-1})\mathbf{E}_v, \tag{9}$$

$$\text{Att}^i(\mathbf{q}_l^i, \mathbf{k}_l^i, \mathbf{v}_l^i) = \text{softmax}(\frac{\mathbf{q}_l^i \mathbf{k}_l^{i^T}}{\sqrt{D_{att}}}\mathbf{v}_l^i), i = 1, 2, \ldots, H \tag{10}$$

$$\mathbf{x}_l' = \text{Concat}(\{\text{Att}^i(\mathbf{q}_l^i, \mathbf{k}_l^i, \mathbf{v}_l^i)\}_{i=1}^H)\mathbf{E}_{out}, \tag{11}$$

where $\mathbf{E}_q$, $\mathbf{E}_k$ and $\mathbf{E}_v \in \mathbb{R}^{D \times D_{att}}$, the dimension of patch embedding is reduced to $D_{att}$ to decrease the computational complexity of Equation (10), $\mathbf{E}_{out} \in \mathbb{R}^{(H \cdot D_{att}) \times D}$, $H$ is the head number of multi-head self-attetion. FTE produces a sequence of encoding features $\{\mathbf{f}_{FTE_i}\}_{i=1}^N$ for each source.

Through the MSA module, each feature point $f_h(i, j)$ on the spatial position $(i, j)$ of the single-source CNN feature performs the global search on the whole feature map and aggregates feature points with similar semantic information. As shown in Equation (10), the cosine distance is used to calculate the semantic similarity matrix between $f_h(i, j)$ and the feature points on each spatial position of $f_h$. The similarity matrix is normalized by the softmax operation. The feature points on $f_h(i, j)$ are weighted and summed using this normalized matrix. Finally, the obtained features are fused with $f_h(i, j)$. FTE makes the features have a larger receptive field; it enhances the discriminative of the single-source features.

**Fusion transformer decoder (FTD).** The FTD is used to integrate multi-source CNN features globally. The multi-source CNN features $\{\mathbf{f}_{FTE_i}^1\}_{i=1}^N$ and $\{\mathbf{f}_{FTE_i}^2\}_{i=1}^N$ are used as the input, and the learnable position embeddings $\{\mathbf{E}_{p_i}\}_{i=1}^N$ are added to input features to make the network sensitive to the spatial information. For each source, FTD is a dual-path transformer, and each path is composed of stacked transformer decoder layers based on MSA (reference Equation (4)), a multi-head cross-attention (MCA) and an FFN. Similarly, skip connection, GELU and LN are used in the decoder. The non-local fusion is performed between the multi-source image features in the multi-head cross-attention layer. $i$ and $j$ are used to mark different sources. For each path, the calculation routine is formulated as

$$\mathbf{y}_0^i = [\mathbf{f}_{FTE_1}^i + \mathbf{E}_{p_1}^i; \mathbf{f}_{FTE_2}^i + \mathbf{E}_{p_2}^i; \cdots; \mathbf{f}_{FTE_N}^i + \mathbf{E}_{p_N}^i], \tag{12}$$

$$\mathbf{y}_l'^i = \text{MSA}(\text{LN}(\mathbf{y}_{l-1})) + \mathbf{y}_{l-1}, \tag{13}$$

$$\mathbf{y}_l''^i = \text{MCA}(\text{LN}(\mathbf{y}_l'^i), \text{LN}(\mathbf{y}_l'^j)) + \mathbf{y}_l'^i, \tag{14}$$

$$\mathbf{y}_l^i = \text{FFN}(\text{LN}(\mathbf{y}_l''^i)) + \mathbf{y}_l''^i, \qquad l = 1 \ldots L \tag{15}$$

$$[\mathbf{f}_{FTD_1}^i, \mathbf{f}_{FTD_2}^i, \ldots, \mathbf{f}_{FTD_N}^i] = \text{LN}(\mathbf{y}_L^i). \tag{16}$$

In Equation (14), similarities between the $n$-th patch embedding of source $i$ and all other patch embeddings of source $j$ are calculated, then similarities are utilized to aggregate all patch embeddings of source $j$ to encode the $n$-th patch embedding. Therefore, MCA is a non-local feature fusion process, which is formulated as follows (for simplicity, only single-head cross-attention is shown):

$$\mathbf{q}_l'^i = \text{LN}(\mathbf{y}_l'^i)\mathbf{E}_q^i, \tag{17}$$

$$\mathbf{k}_l'^j = \text{LN}(\mathbf{y}_l'^j)\mathbf{E}_{k,v}^i, \tag{18}$$

$$\mathbf{v}_l'^j = \text{LN}(\mathbf{y}_l'^j)\mathbf{E}_{k,v}^i, \tag{19}$$

$$\text{Att}'^i(\mathbf{q}_l'^i, \mathbf{k}_l'^j, \mathbf{v}_l'^j) = \text{softmax}(\frac{\mathbf{q}_l'^i \mathbf{k}_l'^{j^T}}{\sqrt{D_{att}}}\mathbf{v}_l'^j), \tag{20}$$

$$\mathbf{y}_l''^i = \text{Att}'^i(\mathbf{q}_l'^i, \mathbf{k}_l'^j, \mathbf{v}_l'^j)\mathbf{E}_{out}^i, \tag{21}$$

where $\mathbf{E}_q^i$, $\mathbf{E}_{k,v}^j \in \mathbb{R}^{D \times D_{att}}$, the dimension of patch embedding is reduced to $D_{att}$ to decrease the computational complexity of Equation (20), $\mathbf{E}_{out}^i \in \mathbb{R}^{D_{att} \times D}$, $i$ and $j$ denote different sources. Finally, FTD generates the decoding feature $\{\mathbf{f}_{FTD}^i\}_{i=1}^N$.

Through the MCA module, each patch embedding $\mathbf{f}_{FTE_n}^i$ of the source $i$ is not directly fused with the patch embedding $\mathbf{f}_{FTE_n}^j$ of source $j$. Instead, the MCA performs a global search on the feature of source $j$. As shown in Equation (20), the cosine distance is used to calculate the semantic similarity matrix between $\mathbf{f}_{FTE_n}^i$ and each patch embedding of source $j$. The similarity matrix is normalized by the softmax operation. The patch embeddings of source $j$ are weighted and summed using this normalized matrix. Finally, the obtained features are fused with $\mathbf{f}_{FTE_n}^i$. Therefore, even if there is the semantic bias between the two sources, FTD fuses the features of source A and those of source B with similar semantic information.

**Semantic class token.** The class token is to perceive global semantic information. Different from patch embedding, the class token is taken as the image representation, which is a learnable embedding. Instead of random initialization, semantic class tokens are utilized to speed up the convergence. Specifically, we perform global average pooling on CNN feature $\mathbf{f}_h \in \mathbb{R}^{H' \times W' \times D}$ to obtain the semantic class token $\mathbf{f}_{sct} \in \mathbb{R}^C$, and $\mathbf{f}_{sct}$ participates in the non-local feature extraction and fusion. To calculate the FTE and FTD, Equations (3), (6), (12) and (16) are modified as follows:

$$\mathbf{x}_0 = [\mathbf{f}_{sct}\mathbf{E}_s + \mathbf{E}_{p_0}; \mathbf{f}_{h_1}\mathbf{E} + \mathbf{E}_{p_1}; \cdots; \mathbf{f}_{h_N}\mathbf{E} + \mathbf{E}_{p_N}], \tag{22}$$

$$[\mathbf{f}_{FTE_0}, \mathbf{f}_{FTE_1}, \ldots, \mathbf{f}_{FTE_N}] = \text{LN}(\mathbf{x}_L), \tag{23}$$

$$\mathbf{y}_0^i = [\mathbf{f}_{FTE_0}^i + \mathbf{E}_{p_0}^i; \mathbf{f}_{FTE_1}^i + \mathbf{E}_{p_1}^i; \cdots; \mathbf{f}_{FTE_N}^i + \mathbf{E}_{p_N}^i], \tag{24}$$

$$[\mathbf{f}_{FTD_0}^i, \mathbf{f}_{FTD_1}^i, \ldots, \mathbf{f}_{FTD_N}^i] = \text{LN}(\mathbf{y}_L^i), \tag{25}$$

where $\mathbf{E}_s \in \mathbb{R}^{C \times D}$. **Feature reorganization.** For IWC and PWC tasks, semantic class tokens of reorganized features are selected from the decoding features and the fusion features $\mathbf{f}_F = \mathbf{f}_{FTD_0}^1 + \mathbf{f}_{FTD_0}^2$. For the SS task, the decoding features are reorganized to restore the original 2D structure: $\{\mathbf{f}_{FTD_i}\}_{i=1}^N \in \mathbb{R}^{N \times D} \overset{reshape}{\longrightarrow} \mathbf{f}_{FTD} \in \mathbb{R}^{\frac{H'}{P} \times \frac{W'}{P} \times D}$.

### 3.2.3. Task Predictor

The task predictor network takes fusion features as the input and produces the prediction map $\tilde{f}_{p_i}$ for each task $T_i$. For different downstream tasks, multiple predictor networks are designed separately. As shown in Figure 4a,b, for PwC and IwC tasks, $\mathcal{F}_{t_i}$ is implemented by fully connected layers. For a semantic segmentation task, $\mathcal{F}_{t_i}$ is implemented by a feature enhancement network (FEN) and a upsample operator. The structure of FEN follows that of the general semantic segmentation model, e.g., the atrous spatial pyramid pooling module in DeepLab v3 and the pyramid pooling module in PSPNet. The size of $\tilde{f}_{p_i}$ depends on the specific task, i.e., $h = H, w = W$ for semantic segmentation task and $h = 1, w = 1$ for PwC and IwC tasks.

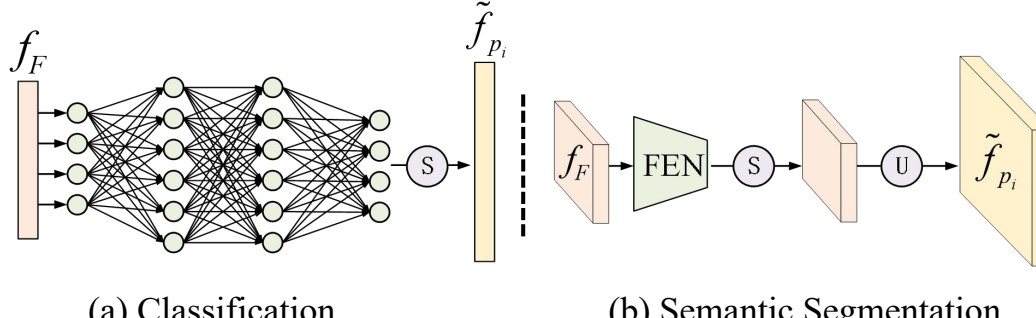

(a) Classification        (b) Semantic Segmentation

**Figure 4.** Task predictor networks. Different task predictor networks are proposed for three classification-based downstream tasks. For (**a**) image- and pixel-wise classification tasks, the predictor is implemented by fully connected layers. For (**b**) semantic segmentation task, the predictor is implemented by a feature enhancement network (FEN) and a upsample operator. 'U' represents the upsample operator and 'S' means soft-max operator.

*3.3. MsIFT Loss*

Let $f_p \in [0,1]^{h \times w \times C}$ be the one-hot style ground-truth map, and fusion features are reorganized and sent to the predictor network to generate score maps $\tilde{f}_p \in [0,1]^{h \times w \times C}$. $C$ is the number of category. The supervised classification loss for training the MsIFT is formulated as follows:

$$\mathcal{L}_{sup} = -\frac{1}{h \times w} \sum_{j=1}^{h} \sum_{k=1}^{w} \sum_{c=1}^{C} f_p^{j,k,c} \log \tilde{f}_p^{j,k,c}. \tag{26}$$

**Auxiliary loss.** To accelerate the training convergence, the predictor network and the classification loss (refer to Equation (26)) are added after the FTE and FTD modules in the training process; all predictor networks share the same parameters. Specifically, the outputs of the FTE and FTD are separately fed into the predictor networks, and the network parameters are additionally trained using the classification loss in Equation (26). The classification losses additionally participating in the training are collectively referred to as auxiliary loss.

The $\alpha$-balanced auxiliary loss $\mathcal{L}_{aux}$ and classification loss $\mathcal{L}_{sup}$ are used as the final loss of the MsIFT: $\mathcal{L} = \mathcal{L}_{sup} + \alpha \mathcal{L}_{aux}$.

## 4. Experiments

*4.1. Data Description*

**VAIS** [9] is a multi-source maritime image dataset for image-wise classification, including 1623 visible images (VIS) and 1242 infrared images (IR), among which there are 1088 corresponding pairs. The dataset contains 6 coarse-grained categories: cargo ship, medium ship, passenger ship, sailing ship, small boat and tug boat. Following the baseline method [9], 1088 VIS/IR image pairs are partitioned into 539 image pairs for training and 549 image pairs for testing. The sample number of each category is listed in Table 1. At the pre-processing stage, each image is resized to $224 \times 224$.

**Table 1.** Number of training and testing samples in VAIS dataset.

| No. | Class Name | Train | Test |
|-----|------------|-------|------|
| 1 | Cargo ship | 83 | 63 |
| 2 | Medium ship | 62 | 76 |
| 3 | Passenger ship | 58 | 59 |
| 4 | Sailing ship | 148 | 136 |
| 5 | Small boat | 158 | 195 |
| 6 | Tug boat | 30 | 20 |
| | Total | 539 | 549 |

The **GRSS-DFC-2013 dataset** [11] is a data fusion dataset for pixel-wise classification, which is composed of HSIs and LiDAR data. HSIs have 144 spectral bands from 380 nm to 1050 nm. The LiDAR data have one band; the spatial resolution of both HSIs and LiDAR data is 2.5 m/pixel. GRSS-DFC-2013 was acquired over the University of Houston, which contains 15 categories. As shown in Table 2, the sizes of the training set and test set are the same as [14]. The technique in [14] is used for image pre-processing. For each pixel, 7-neighborhood pixels are sampled as the input, i.e., the input size is $7 \times 7$.

**Table 2.** Number of training and testing samples in GRSS-DFC-2013 dataset.

| No. | Class Name | Train | Test |
|---|---|---|---|
| 1 | Health grass | 198 | 1053 |
| 2 | Stressed grass | 190 | 1064 |
| 3 | Synthetic grass | 192 | 505 |
| 4 | Tress | 188 | 1056 |
| 5 | Soil | 186 | 1056 |
| 6 | Water | 182 | 143 |
| 7 | Residential | 196 | 1072 |
| 8 | Commercial | 191 | 1053 |
| 9 | Road | 193 | 1059 |
| 10 | Highway | 191 | 1036 |
| 11 | Railway | 181 | 1054 |
| 12 | Parking lot 1 | 192 | 1041 |
| 13 | Parking lot 2 | 184 | 285 |
| 14 | Tennis court | 181 | 247 |
| 15 | Running track | 187 | 473 |
| | Total | 2832 | 12,197 |

**SpaceNet 6** [10] is a multi-source remote sensing image dataset for semantic segmentation, and it is composed of optical images (OPT) and SAR imagery over the port of Rotterdam, The Netherlands. The SAR images are provided by Capella Space, and each image has four polarizations (HH, HV, VH and VV). The optical images are captured by the Maxar Worldview-2 satellite. Three sets of optical data with different spatial resolutions are provided: 0.5 m/pixel panchromatic image, 2.0 m/pixel multi-spectral images of four bands (blue, green, red and near-infrared) and 0.5 m/pixel pansharpened multi-spectral image (blue, green, red and near-infrared). Pansharpened images and SAR images are selected to evaluate the performance of multi-source image semantic segmentation. The semantic segmentation annotations corresponding to the SAR image are used as the ground truth. A total of 3401 unregistered optical–SAR image pairs are partitioned into 1700 pairs for training and 1701 pairs for testing. Each image is resized to $900 \times 900$.

### 4.2. Implementation Details

The proposed MsIFT is implemented based on the open-source computer vision toolboxes MMClassification [34] and MMSegmentation [35]. The experiments were conducted on a server cluster with a 64-bit Linux operating system. The hardware includes Tesla V100 GPU (32 GB memory) and Intel(R) Xeon(R) Gold 6230 CPU @ 2.10 GHz.

The model pre-trained on ImageNet [36] was used to initialize the CNN feature extractor network. The other training settings are listed in Table 3. SGD denotes stochastic gradient descent.

**Table 3.** Training settings on different downstream tasks.

| | Pixel-Wise Classification | Image-Wise Classification | Semantic Segmentation |
|---|---|---|---|
| Batchsize | 48 | 10 | 4 |
| Optimizer | SGD | SGD | SGD |
| Initialized learning rate | 0.001 | 0.001 | 0.01 |
| Learning Rate Decay | Cosine Annealing | Cosine Annealing | Poly schedule |
| Momentum | 0.9 | 0.9 | 0.9 |
| Weight decay | 0.0001 | 0.0001 | 0.0005 |
| Epochs | 600 | 200 | 30 |

*4.3. Evaluation Metrics*

Overall accuracy (*OA*) is used to quantify the pixel- and image-wise classification performance, which is formulated as follows:

$$OA = \frac{\sum_{i=1}^{C} N_{ii}}{\sum_{i=1}^{C} \sum_{j=1}^{C} N_{ij}}, \tag{27}$$

where $C$ denotes the number of categories. $N_{ij}$ represents the number of samples that belong to class $i$ but are predicted to be class $j$ and $N_{ii}$ is the number of samples being correctly classified.

Pixel accuracy (*PA*) and mean intersection over union (*mIoU*) are used to quantify the segmentation performance, and they are formulated as follows:

$$PA = \frac{\sum_{i=1}^{C} p_{ii}}{\sum_{i=1}^{C} \sum_{j=1}^{C} p_{ij}}, \tag{28}$$

$$mIoU = \frac{1}{C} \sum_{i=1}^{C} \frac{p_{ii}}{\sum_{j=1}^{C} p_{ij} + \sum_{j=1}^{C} p_{ji} - p_{ii}}, \tag{29}$$

where $C$ denotes the number of categories, $p_{ij}$ denotes the number of pixels that belong to class $i$ but are predicted to be class $j$.

*4.4. Quantitative Analysis*

The proposed MsIFT is evaluated on three datasets for image classification, pixel-wise classification and semantic segmentation tasks. Ablation experiments are aimed to verify the effectiveness of the components in the MsIFT.

4.4.1. Image-Wise Classification

Classification performances of the MsIFT and the baseline methods on VAIS are listed in Table 4. The multi-source image fusion methods for image-level classification that have achieved SOTA performance on this dataset are selected as the baseline methods. All the methods are conducted using daytime images. In this experiment, ResNet-50 is used as the backbone network. The classification accuracy on VIS and SAR images are 87.1% and 72.1%, respectively. It can be informed that the selected backbone network has no better performance than the single-source classification baseline model, even lower than SF-SRDA. Moreover, because IR images are inferior to VIS images in color and texture, etc., the classification accuracy of VIS images is 15% higher than that of IR images. In terms of multi-source image fusion classification performance, the MsIFT improves the classification accuracy to 92.3%. After merging features of VIS and IR images, compared with ResNet-50, the MsIFT improves the classification accuracy by 5.2% and 20.2%, respectively, which is better than all baseline methods. The MsIFT achieved SOTA performance even if the single-source classification model had no leading performances, which shows the solid global feature extraction and fusion performance of the proposed MsIFT.

**Table 4.** Performance comparison on VAIS dataset.

| Method | VIS | IR | VIS + IR |
|---|---|---|---|
| CNN [9] | 81.9 | 54.0 | 82.1 |
| Gnostic field [9] | 82.4 | 58.7 | 82.4 |
| CNN + gnostic field [9] | 81.0 | 56.8 | 87.4 |
| ME-CNN [37] | 87.3 | - | - |
| MFL (feature-level) + ELM [38] | 87.6 | - | - |
| CNN + Gabor + MS-CLBP [38] | 88.0 | - | - |
| Multimodal CNN [12] | - | - | 86.7 |
| DyFusion [13] | - | - | 88.4 |
| SF-SRDA [39] | 87.6 | 74.7 | 88.0 |
| MCFF Combination 3-SUM (C2C5F6) [2] | 87.5 | 71.1 | 89.6 |
| MCFF Combination 3-SUM (C3C5F6) [2] | 87.7 | 71.4 | 89.6 |
| MCFF Combination 2-CON (C3C5F6) [2] | 87.9 | 71.9 | 89.9 |
| MCFF Combination 3-CON (C2C5F6) [2] | 87.5 | 71.1 | 89.9 |
| MsIFT (ours) | 87.1 | 72.1 | **92.3** |

### 4.4.2. Pixel-Wise Classification

In the experiments, the representative methods of traditional methods [40], CNN [1,14], capsule network [4] and LSTM network [3,21] are used as baseline methods to demonstrate the advantages of the proposed transformer-based multi-source fusion network. Table 5 lists the performance of the MsIFT on pixel-wise classification. The backbone network used to extract local features of multi-source images is consistent with MDL. MDL enhances the feature fusion performance based on CNN. The MsIFT improves the baseline method MDL by 1.03%, and it outperforms all baseline methods. At the same time, the MsIFT improves the performance of single-source image classification to the greatest extent (i.e., 10.97%).

**Table 5.** Performance comparison on HS–LIDAR dataset.

| Method | HSI | LiDAR | HSI + LiDAR |
|---|---|---|---|
| SVM [40] | 78.79 | - | 80.15 [+1.36] |
| ELM [40] | 79.52 | - | 80.76 [+1.24] |
| Two-Branch CNN [1] | 77.79 | - | 83.75 [+5.96] |
| Dual-Channel CapsNet [4] | 81.53 | - | 86.61 [+5.08] |
| SSCL3DNN [3] | 82.72 | - | 86.01 [+3.29] |
| $A^3$CLNN [21] | 87.00 | - | 90.55 [+3.55] |
| MDL + Early [14] | 82.05 | 67.35 | 83.07 [+1.02] |
| MDL + Middle [14] | 82.05 | 67.35 | 89.55 [+7.5] |
| MDL + Late [14] | 82.05 | 67.35 | 87.98 [+5.93] |
| MDL+ EnDe [14] | 82.05 | 67.35 | 90.71 [+8.66] |
| MDL + Cross [14] | 82.05 | 67.35 | 91.99 [+9.94] |
| MsIFT (ours) | 82.05 | 67.35 | **93.02 [+10.97]** |

### 4.4.3. Semantic Segmentation

DeconvNet-Fusion [24] is chosen as the baseline method for semantic segmentation based on multi-source images fusion. Two voting mechanisms, arithmetic mean (*AM*) and geometric mean (*GM*), are introduced to extend the method used for fusion in DeconvNet-Fusion:

$$M_f(i,j)^{AM} = \frac{M_{SAR}(i,j) + M_{OPT}(i,j)}{2},$$
$$M_f(i,j)^{GM} = \sqrt{M_{SAR}(i,j) \times M_{OPT}(i,j)}, \tag{30}$$

where $M_f$ is the fused saliency map defined in [24].

DeconvNet-Fusion used the fully convolution network (FCN) as the single-source image segmentation model. In order to demonstrate the generalization performance of the fusion module on different semantic segmentation models, more SOTA segmentation

models, PSPNet [41] and DANet [42], are chosen as the single-source segmentation models in MsIFT and DeconvNet-Fusion. Deeplabv3+ [43], OCRNet [44] and CCNet [45] are used as baseline models. It is worth noting that the proposed MsIFT is compatible with all single-source segmentation models.

As shown in Table 6, DANet achieves the best performance among single-source image segmentation models, where mIoU in OPT images is 66.90%, and the accuracy is 70.17%. The mIoU in SAR images is 57.45%, and the accuracy is 62.64%. Regarding multi-source image segmentation performance, although DeconvNet-Fusion is better than SAR image segmentation performance, it is lower than OPT image segmentation performance. The reason is that the strategy of corresponding position fusion used in DeconvNet-Fusion is limited for unregistered datasets. The fusion module of MsIFT has global receptive fields for features from different sources, which effectively reduced the semantic bias caused by unregistered images. Finally, both mIoU and the accuracy of multi-source image segmentation exceed that of single-source image segmentation models. Specifically, for PSPNet, the MsIFT improves mIoU on OPT images by 1.69%, and the accuracy increased by 1.64%. For DANet, the MsIFT increased mIoU on OPT images by 1.04% and the accuracy by 0.65%.

**Table 6.** Performance comparison on SpaceNet6.

| Method | Backbone | OPT | | SAR | |
| --- | --- | --- | --- | --- | --- |
| | | **mIoU** | **Accuracy** | **mIoU** | **Accuracy** |
| Deeplabv3+ [43] | ResNet-50 | 63.36 | 67.18 | 56.46 | 61.87 |
| OCRNet [44] | HRNetV2-W18 | 65.74 | 68.58 | 54.56 | 59.65 |
| CCNet [45] | ResNet-50 | 65.49 | 68.82 | 55.36 | 60.61 |
| PSPNet [41] | ResNet-50 | 65.82 | 68.85 | 55.32 | 60.54 |
| DANet [42] | ResNet-50 | 66.90 | 70.17 | 57.45 | 62.64 |
| **Fusion Method** | **Backbone** | OPT + SAR | | | |
| | | **mIoU** | | **Accuracy** | |
| *PSPNet* [41]: | | | | | |
| DeconvNet-Fusion (Minimum) [24] | ResNet-50 | 58.25 | | 59.65 | |
| DeconvNet-Fusion (AM) [24] | ResNet-50 | 64.06 | | 68.83 | |
| DeconvNet-Fusion (GM) [24] | ResNet-50 | 54.92 | | 66.28 | |
| MsIFT (Ours) | ResNet-50 | **67.51** | | **70.49** | |
| *DANet* [42]: | | | | | |
| DeconvNet-Fusion (Minimum) [24] | ResNet-50 | 60.03 | | 61.47 | |
| DeconvNet-Fusion (AM) [24] | ResNet-50 | 65.09 | | 70.01 | |
| DeconvNet-Fusion (GM) [24] | ResNet-50 | 56.91 | | 66.2 | |
| MsIFT (Ours) | ResNet-50 | **67.94** | | **70.82** | |

### 4.4.4. Ablation Study

Semantic segmentation is used as a downstream task in ablation study to justify the performance of three critical components in the MsIFT: fusion transformer encoder, fusion transformer decoder and auxiliary loss (AL). It is worth noting that when only the FTE is used, the fusion features are obtained by adding the encoding features from two sources obtained by the FTE.

From Table 7, the auxiliary loss enhanced the training of the FTE in the MsIFT and finally improved the fusion performance. After using AL, mIoU is increase by 1.32. From the ablation experiments on the FTE and FTD, it can be inferred that the FTD obtained the higher fusion performance than the FTE (66.08 vs. 65.95). Compared with CNN, even if the FTE has the global receptive field in single-source feature extraction, the direct addition of corresponding pixel-wise features is local fusion, and the FTD module has the global receptive field in the multi-source feature fusion procedure. Therefore, adding the FTD module helps the MsIFT achieve SOTA performance.

To investigate how the fusion module of the MsIFT works, two other local fusion methods are selected as baseline methods: CNN and Concat, which directly replace the FTE and FTD modules.

**Table 7.** Ablation study on MsIFT.

| OPT | SAR | Concat | CNN | FTE | FTD | AL | mIoU |
|:---:|:---:|:------:|:---:|:---:|:---:|:--:|:----:|
| ✓ | | | | | | | 64.07 |
| | ✓ | | | | | | 54.28 |
| ✓ | ✓ | ✓ | | | | | 50.21 |
| ✓ | ✓ | | ✓ | | | | 48.53 |
| ✓ | ✓ | | | ✓ | | ✓ | 65.95 |
| ✓ | ✓ | | | | ✓ | ✓ | 66.08 |
| ✓ | ✓ | | | ✓ | ✓ | | 65.06 |
| ✓ | ✓ | | | ✓ | ✓ | ✓ | **66.38** |

**Concat**: Directly concatenate CNN features from multi-sources along the channel dimension.

**CNN**: Based on the above concatenate operation, the convolution layer is added to modify the fusion features locally.

It can be observed that these two methods fail in outperforming the single-source segmentation model with respect to mIoU and accuracy, which illustrates that the fusion method without considering the global receptive field is limited. However, compared with the segmentation performance on OPT images, the MsIFT effectively improves mIoU by 2.31%.

### 4.5. Qualitative Analysis

**How can the MsIFT handle semantic bias?**: Figure 5a,c shows attention maps in which the query points on one source image aggregates features on another source image during the cross-source fusion process. The highlighted position represents the area concerned by the query points. Compared with simple pixel-wise multi-source feature fusion, **the MsIFT is more powerful in helping the query point conduct the global search on other source features and aggregate features with similar semantic information to the query point features.** The yellow dot represents the foreground query point. As can be induced from its corresponding attention map, the foreground query point focuses on the building in another source image. The background query point focuses on the background in another source image. Therefore, at the cross-source fusion stage, the MsIFT semantically aligns the features from different source images through the attention mechanism and avoids the semantic bias caused by misregistration. This is the reason why the MsIFT is superior to other fusion methods. Figure 5b,d shows the location that query points focus on when aggregating features from the single-source feature encoder. Similarly, each feature selects contextual features with similar semantic information and has a global receptive field. Therefore, it further improves the discriminability of features.

**Fusion results visualization**: Figure 6 shows the fusion results on the semantic segmentation of multi-source remote sensing images. Because the imaging mechanisms of the SAR image and the optical image are inconsistent, the semantic bias is more negative. Figure 6c,d shows the results on the SAR and optical images, respectively. Since the texture and color information are more rich in the optical image, the segmentation result on the optical image is more accurate. Figure 6b shows the results of the MsIFT. The zoomed area (red dashed box) shows the advantages of the MsIFT. Interestingly, by multi-source image feature fusion and joint decision segmentation, the MsIFT corrected the regions misclassified by the single-source image segmentation model. In a word, the MsIFT successfully improves the segmentation performance.

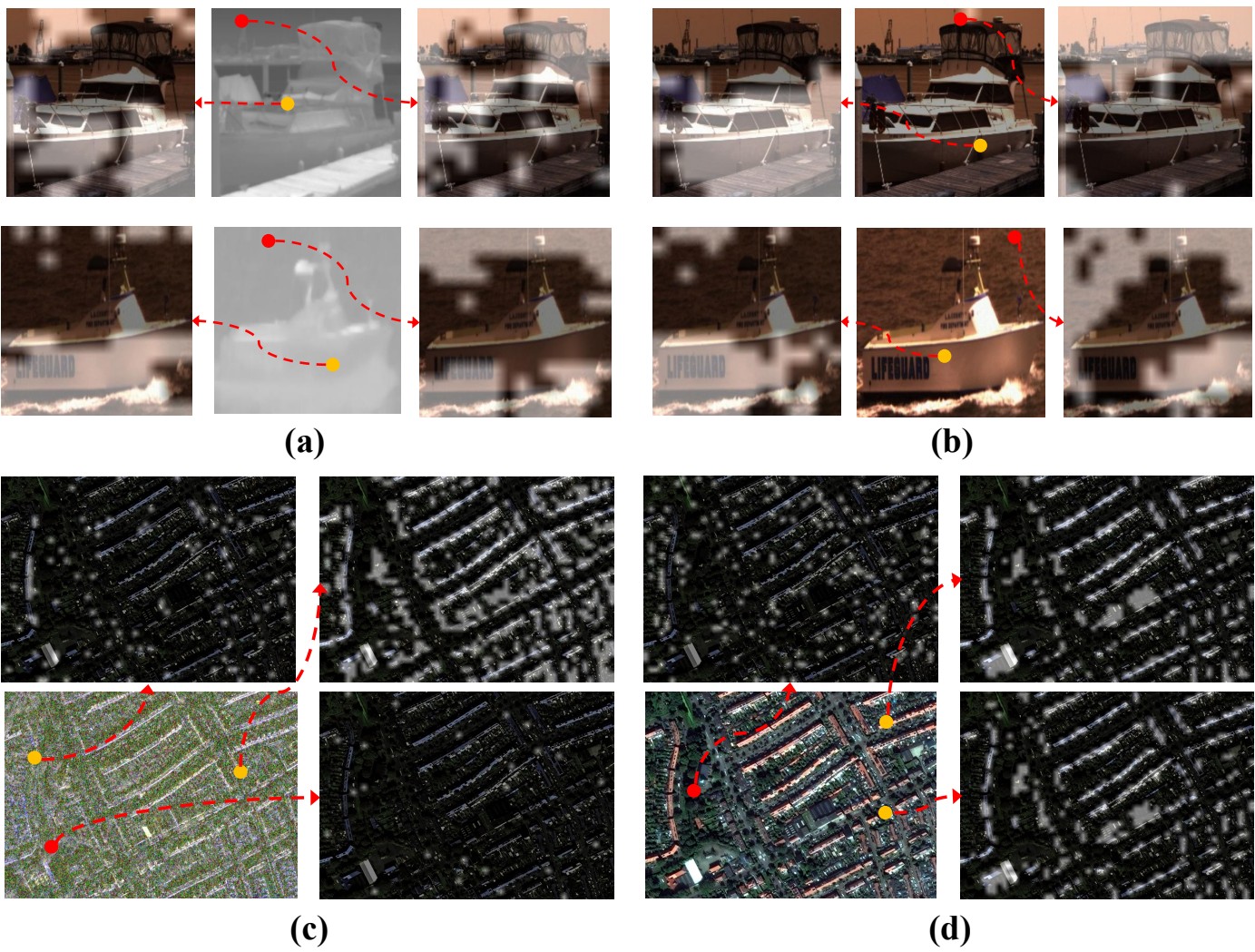

**Figure 5.** Attention maps of the picked points. (**a**–**d**) are the attention maps from classification and semantic segmentation, respectively. (**a**,**c**) are multi-source cross-attention maps. (**b**,**d**) are single-source self-attention maps. The figures indicated by the arrow show the global attention maps of the picked query point. The red points represent the background queries, and the yellow points are the foreground queries.

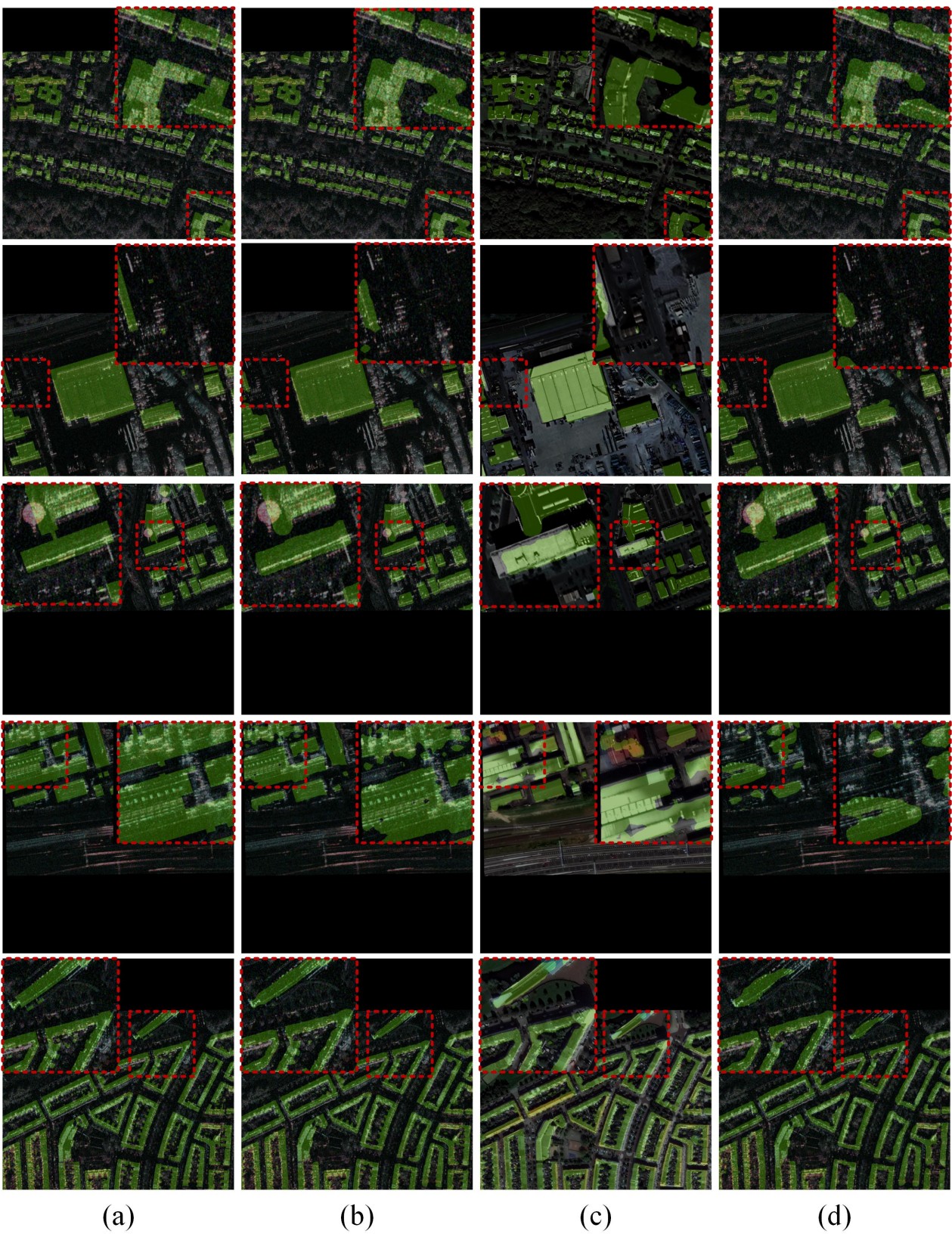

(a)          (b)          (c)          (d)

**Figure 6.** Results comparison of multi-source semantic segmentation, (**a**) ground truth, (**b**) MsIFT, (**c**) OPT and (**d**) SAR. Different rows represent the results in different scenarios. The red dotted rectangle areas are enlarged to show the results more clearly.

## 5. Conclusions

A novel image fusion framework, MsIFT, is proposed in this paper for multi-source image fusion. The MsIFT integrates three downstream classification tasks: i.e., pixel-wise classification, image-wise classification and semantic segmentation. Different task-specific networks are designed for local feature extraction and prediction, respectively. Three tasks share the multi-source features fusion module within the MsIFT. A feature fusion transformer (FFT) with encoder–decoder style is proposed for multi-source feature-level fusion; the global attention mechanism is beneficial for alleviating semantic biases caused by inaccurate registration. The FFT allows features to perform global queries, inspiring each query feature to aggregate global features similar to their semantic information. Extensive experiments demonstrate that the MsIFT achieved state-of-the-art performances on VAIS, GRSS-DFC-2013 and SpaceNet 6, which validates the superiority and versatility of the proposed method. In the future work, we will extend the MsIFT framework to more downstream tasks.

**Author Contributions:** Conceptualization, X.Z. and C.H.; methodology, X.Z., H.J. and C.H.; software, X.Z.; validation, X.Z.; formal analysis, H.J.; investigation, H.J., N.X. and L.N.; data curation, X.Z.; writing—original draft preparation, X.Z.; writing—review and editing, L.N. and C.H.; visualization, X.Z.; supervision, C.H. and C.P.; funding acquisition, C.H. All authors have read and agreed to the published version of the manuscript.

**Funding:** This research is supported by National Natural Science Foundation of China (Grant No. 62071466), Fund of National Key Laboratory of Science and Technology on Remote Sensing Information and Imagery Analysis, Beijing Research Institute of Uranium Geology (Grant No. 6142A010402) and Guangxi Natural Science Foundation (Grand No. 2018GXNSFBA281086).

**Data Availability Statement:** Three public datasets (i.e., VAIS, GRSS-DFC-2013, and SpaceNet6) were analyzed in this study. The data of VAIS was downloaded from the official website: https://chriskanan.com/datasets/ (accessed on 2015). The data of GRSS-DFC-2013 was downloaded from the official website: https://hyperspectral.ee.uh.edu/?page_id=459 (accessed on 2013). The data of SpaceNet6 was downloaded from the official website: https://spacenet.ai/sn6-challenge/ (accessed on 2020).

**Conflicts of Interest:** The authors declare no conflict of interest.

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
