# Peer review of "MsIFT: Multi-Source Image Fusion Transformer"

_remotesensing, doi:10.3390/rs14164062_

Round 1
Reviewer 1 Report
This paper proposes a novel multi-source image fusion transformer (MsIFT) model for vision tasks. Semantic bias caused by misregistration is an impediment to image fusion and the authors aim to solve this problem. The research is meaningful and the work is substantial. However, some issues need to be clarified.
## Introduction
1. The introduction or related works part needs more content about image fusion and the development of related methods to make readers understand your contributions.
## Related works
2. In line 112 and 113, the sentence ‘ViT has successfully outperformed CNN in recognition, which has dramatically 113 stimulated researchers’ enthusiasm to introducing the transformer into the computer vision world.’ is too absolute because CNN is a broad concept. If it is true, please add citations.
3. Although this part has introduced a lot about computer vision, the contents seem a little deviation from the title “multi-source image fusion”. In addition, this part fails to exhibit the latest developments, the research hotspots and the difficulties in the field.
## Methods
4. In my understanding, section 3.2.2 may be the most important part in methods, but it’s hard for me to understand how this module works. In other words, can you explain why your module can solve semantic bias or what are the advantages from the aspect of the theory?
5. More explanations are needed for auxiliary loss.
## Experiments
6. So many methods were chosen in this part. You should explain why they are selected for comparison and if they can demonstrate the innovation of your method.
7. In addition, if you aim to address the problem caused by misregistration, can you exhibit some samples with this problem in the datasets?
8. Why semantic segmentation task is used for ablation study while others are not?
9. I’m sorry that it is a little hard for me to understand the meaning of figure 5. Can you explain that?
Reviewer 2 Report
This paper presents a method for exploiting the vision transformer to address the semantic bias problem in multi-source image fusion. After scrutinizing the manuscript, the reviewer has some comments as follows.
1. The vision transformer should be introduced in considerable detail because it is the heart of the proposed method. The authors are highly suggested to describe how the transformer models long-range dependence through the attention mechanism. If possible, please utilize mathematical expressions and visual illustrations for an intuitive description.
2. What does the minus sign on the right side the MCA block in Figure 2 mean? Its description should be added to the caption of Figure 2.
3. The description of the feature fusion transformer is quite limited, causing difficulties to access the paper’s novelties. Thus, it is advisable to provide more details about the proposed transformer, notably the MCA and the MSA.
4. It is suggested to double-check and revise (if necessary) equations and notations in the paper for correctness and consistency. For example, the reviewer could not find definitions of Datt in Equation (10) and Laux on line 206. Also, as the number of patches N is an integer, its calculation should be protected by a rounding, flooring, or ceiling operator.
5. According to the results of SF-SRDA [30], it appears that the performance comparison in Table 4 was conducted using daytime images. If so, please revise the description in the main text for clarity.
Round 2
Reviewer 1 Report
Accept.